# Developing and testing the feasibility of a new internet-based intervention–A case study of people with stroke and occupational therapists

Ida-Maria Barcheus[1]*, Maria Ranner[1‡], Anneli Nyman[1‡], Eva Månsson Lexell[2,3‡], Maria Larsson-Lund[1]

1 Department of Health, Education and Technology, Division of Health, Medicine and Rehabilitation, Luleå University of Technology, Luleå, Sweden, 2 Department of Health Sciences, Lund University, Lund, Sweden, 3 Department of Neurology, Rehabilitation Medicine, Memory Clinic and Geriatrics, Skåne University Hospital, Lund-Malmö, Sweden

☯ These authors contributed equally to this work.
‡ MR, AN and EML also contributed equally to this work.
* Ida.maria.barcheus@ltu.se

**Data Availability Statement:** The data underlying this study cannot be made publicly available in

## Abstract

### Introduction

Internet-based interventions are called for within rehabilitation to meet the limited access to support for self-management after stroke. Therefore, a new intervention program, "Strategies for Empowering activities in Everyday life" (SEE) was developed. The aim of this study was to explore and describe how clients with stroke and their occupational therapists experienced the SEE intervention process and whether SEE has the potential to promote an active everyday life.

### Methods

A qualitative descriptive case study was designed. Four people with stroke (two of each sex, mean age 66,5 years) and their two occupational therapists (one of each sex) were included. A mix of data collection methods as interviews, assessments, registration forms and fieldnotes was used to uncover the participants' experiences and potential changes. Data were analysed with pattern matching.

### Findings

The analysed data formed three categories: *"Not being able to take on the internet-based intervention"*, *"Being facilitated in the change process of everyday life through the internet-based intervention"*, and *"Providing a new internet-based intervention is a transition from ordinary practice"*. These categories included two to four subcategories that reflected aspects of SEE feasibility and acceptability with a focus on content and delivery.

open databases due to ethical and legal reasons, according to Swedish law and regulations. Further, the ethical approval for the study does not admit that data are openly shared. The ethical application and approval for the study can be requested from the Swedish Ethical Review Authority (Dnr: 2019-04993). Data are archived at Luleå University of Technology (LTU) in accordance with Swedish law and regulations and interested researchers may only obtain data after receiving approval from the Swedish Ethical Review Authority. Requests to access data can be sent to the data protection office at LTU, registrator@ltu.se No contact with authors is necessary to request data exchange.

**Funding:** This study was supported by grants from the Kamprad Family Foundation for Entrepreneurship, Research & Charity. The funder had no role in the study.

**Competing interests:** The authors have confirmed that no competing interests exist.

## Conclusion

The first test of the intervention indicates that the content and delivery of SEE can be feasible and acceptable both for clients and occupational therapists. The findings suggest that SEE has the potential to support clients' self-reflections and their adoption of strategies that influence engagement in daily activities and satisfaction with life in various ways. Further research with large-scale studies is needed.

## Introduction

Stroke is the third-leading cause of long-term disability in the adult population worldwide, often leading to difficulties with returning to a previous way of living life [1]. Many people with stroke find it challenging to engage in daily activities, especially in places outside the home, affecting their social contentedness and sense of belonging. This can negatively influence their patterns of daily activities in an ordinary day or week, that is, how, where and with whom they occupy their time [2]. In turn, the possibility to achieve a healthy occupational balance [3, 4], defined as a subjective experience of having a satisfying amount and variation of daily activities, becomes disrupted. A person with stroke who experiences an unhealthy occupational balance may also experience less value and meaning in daily activities, which is important for health and life satisfaction [5]. This often become obvious after discharge from the hospital, when the person no longer has access to health care poststroke and yet faces persistent challenges in everyday life [6, 7].

Rehabilitation that support self-management [8]–when clients are provided knowledge and skills to actively manage everyday life and facilitate their adaption to the consequences of stroke is therefore needed [7, 9–11]. However, access to rehabilitation in the later phases of stroke, is insufficient and unequally geographically distributed [12, 13]. Unfilled needs in daily activities are also common since the rehabilitation offered does not often specifically focus on facilitating an active everyday life [7, 13, 14]. In line with this, the action plan for stroke in Europe [15] emphasizes the need to develop services targeting life after stroke. An improved support to help people self-manage long-term issues to come to terms with their new situation is requested. In this, it is suggested that new modes of services need to be used as resource. Reviews [16–18] show a growing use of digital components in parts of the rehabilitation process after stroke, commonly for assessment or training. However, the use of internet-based solutions in supporting self-management of activities in everyday life is sparse. Thus, to complement the rehabilitation provided at rehabilitation facilities, new self-management programs that take advantage of the potential of new modes of delivery are needed. For instance, internet-based services can remove transportation barriers and reduce the personal effort and stress related to accessing rehabilitation facilities. Receiving internet-based services can increase motivation and enhance the implementation of flexible individually tailored rehabilitation anchored in everyday life [19–23]. To meet these needs, the internet-based intervention "Strategies for Empowering activities in Everyday life" (SEE, version 1.0) [14] was developed.

SEE [14] is developed for people with neurological conditions, such as stroke, who need support during the change process they go through when adapting to their new life situations. The content of SEE is designed to enable clients to "see" their daily activities in a new light by self-reflection and to develop self-initiated management strategies in the activities. The goal is thereby to empower clients to take an active role in preventing and overcoming problems and challenges in everyday life in a way that becomes sustainable over time [14]. The development

of SEE follows the Medical Research Council guidance (MRC) for how complex interventions are developed and evaluated [24]. Prior stages of the SEE development phase involved reviewing and developing evidence to underpin the intervention and modelling the prototype in collaboration with stakeholders [14]. The development phase of SEE has now reached the stage of initially testing the prototype in clinical practice. As the intervention is new (both in terms of delivery and content), it is important to identify possible alterations and improvements before a larger-scale feasibility study is performed. Testing SEE on a few cases, closely following both clients and clinical occupational therapists (OTs) during pre- and postintervention, can provide important knowledge of SEE's (i) feasibility and acceptability, with a focus on delivery and content, as well as (ii) experienced changes and outcomes.

### Aim

The aim of this study was to explore and describe how clients with stroke and their OTs experience the SEE intervention process and whether SEE has the potential to promote an active everyday life.

## Material and methods

### Design

An exploratory descriptive case study design [25] was employed to provide rich in-depth descriptions of individuals in their real context i.e., clients and occupational therapist in their clinical context. As this was the initial test of SEE, it was favorable to include both groups of users (those who received or delivered SEE) in the case study to provide for a comprehensive understanding of different perspectives. Several data collection methods were employed before, during and after the SEE intervention. The study was reviewed and approved by the Swedish Ethical Review Authority (Dnr: 2019–04993).

### Recruitment

A consecutive selection of potential participants was performed from a list of patients who had received early supported discharge and rehabilitation in the home from a hospital in the northern part of Sweden according to the inclusion criteria: (i) stroke onset > 6 months, (ii) aged 18–75 years, (iii) moderate disability or good recovery based on the extended Glasgow Outcome Scale [26], (iv) able to express themselves verbally and in writing, (v) access to the internet (computer, tablet, or phone they were able to use), and (vi) be motivated to change activities in everyday life. Their motivation to change was explored by a set of questions [27] investigating if daily activities were a problem and if there were a readiness to implement a change. The person needed to respond positively on the questions to be included. The exclusion criteria were (vii) other illnesses/injuries affecting daily activities and (viii) depression based on Hospital Anxiety and Depression scale [28]. The client participants inclusion included a three steps process, as the information at the clinic did not provide updated information on all criteria. Therefore, supplementary assessment needed to be conducted by the researchers (step two and three).

In the first step of the inclusion process, an OT at the clinic identified potential participants based on criteria (i-iv, vii) Research information letters were sent to potential participants, including information about the intervention and that participation was voluntary. Those who were interested in participating were instructed to send a written response to the clinic. Those who did not respond were contacted by the OT by telephone to ensure that no potential participants were missed. Contact information for those who were interested was forwarded to the

**Table 1. Characteristics of the client with stroke receiving SEE, n = 4.**

| | |
|---|---|
| Female/Male | 2/2 |
| Age, Mean (range) | 66, 5 (59–69) |
| Time since injury, months, mean (range) | 7 (6–9) |
| Severity of disability[1] good recovery/moderate disability | 3/1 |
| Prescence of anxiety/depression[2] | 0 |
| Mental Fatigue Scale, Mean (range) [3] | 11 (10–12) |
| Education level: Upper secondary school/ University | 3/1 |
| Working part-time/Retired | 1/3 |
| Living condition: | |
| Married/cohabiting and children not living at home | 4 |
| Lives in own house | 4 |
| Driving car, yes/no | 3/1 |
| Previous received rehabilitation after in-patient care | |
| Follow-up calls | 1 |
| Primary care, physiotherapy | 2 |
| Out-patient rehabilitation | 1 |

[1] Severity of disability according to Glasgow Outcome Scale (GOSE) ratings: Severe disability, Moderate disability, Good recovery.

[2] Hospital Anxiety and Depression Scale (HAD).

[3] MFS, min-max score: 0–44 p, with cut-off score of 10.5 where fatigue should be considered/affects activity

research team. In the second step of the inclusion process, the potential participants were contacted by the research team (IMB, MR) by phone to screen for inclusion criteria (iii, v). In a third and final step of the inclusion process, a last screening (criteria vi, viii) was conducted in a video meeting. Throughout the inclusion process, besides the written information letters, potential client participants received verbal information about the research and their possibilities to withdraw at any time. Those who were assessed eligible to participate were informed again before their verbal and written informed consent was sought. All client participants agreed in written to participate.

In all, twenty-one potential participants with stroke were asked to participate. Four agreed to participate, two could not be reached and fifteen declined. The four clients were between the ages of 59 and 69 years (Table 1).

Two OTs at the rehabilitation clinic were also recruited to provide SEE to the clients. The OTs received verbal information about the study and an information letter before they provided their verbal and written informed consent to participate. The OTs (one man and one woman) had ten and twenty-four years of experience in the profession and had substantial clinical experience working with clients with stroke.

## Intervention

The internet-based occupational therapy intervention SEE [14] aims to support a balanced pattern and level of engagement in various daily activities in different places and with other people to promote an active everyday life and health. SEE is a person-centred intervention and is designed to support clients in developing self-management strategies that empower engagement in daily activities. An in-depth description of the development and the knowledgebase for SEE has been presented previously [14]. SEE is available through the "support and treatment platform" within the Swedish 1177 healthcare guide, the national platform for web-based

healthcare services [29]. Before the start of the intervention, the clients received guides with information about SEE content and structure and how to use the 1177 support and treatment platform. The intervention process starts with a structured evaluation phase including assessments with various tools and a dialogue between the client and the OT, supporting clients to begin their self-reflection of activities in everyday life and to use self-initiated management strategies in activities [14]. This evaluation phase, used as a therapeutic means in SEE and as an occasion for collecting baseline data, was conducted by the researchers (IMB, MR). The results of the evaluation phase were, with the consent of the clients, verbally reported to the OT who provided SEE. Thereafter, the client was introduced to SEE by the OT in the 1177 support and treatment platform during a video meeting. A two-to-three-week intense web program followed, which consisted of eight modules focusing on motivation for change, daily activities, and management strategies. Each module has its unique theme: i) introduction to SEE, ii) changes in activities of everyday life after stroke, iii) the value and meaning of activities, iv) activity pattern, v) the complexity of activities, vi) activity balance, vii) self-initiated strategies to manage activities, and viii) prioritization of changes in activities. The eight modules include short educational video clips that are further processed through digital assignments, digital feedback and three online face-to-face guiding sessions with the OT [14]. The web program is intended to result in an activity plan that the client establishes together with the OT. The activity plan includes an overall goal and subgoals with associated management strategies for activities and a time plan for evaluation. Thereafter, they work together, in a person-centred manner, with close follow-ups, to implement the activity plan until goals are reached.

To prepare the OT to implement SEE, they participated in a standardized web-based educational program about its evidence base and program theory and were educated on how to use the platform. They were also supported by an intervention guide with detailed information about how to support client processes through each module, and they had the possibility to receive supervision from the researchers (IMB, MLL, MR) whenever needed.

## Data collection and procedure

To follow the intervention process and evaluate the feasibility of SEE, data were collected with several methods on four occasions (i-iv) (Table 2). In the evaluation phase/baseline **(i)**, standardized assessment tools were used to evaluate the complexity of an active everyday life and changes herein to support and evaluate the outcome of the process. The primary outcome assessments focused on aspects of activities in everyday life and how these had been managed based on both self-reports and professional assessment. These were an interview instrument based on a time-use diary—the profiles of occupational engagement (POES) [30, 31]—and self-reports using the occupational balance questionnaire (OBQ11) [32, 33] and the occupational values with predefined items (Oval-pd) [34]. The secondary outcome assessments included additional aspects that further can be affected and contribute to interpretation of the potential results. These were the life satisfaction questionnaire (Lisat-11) [35], the work ability index (WAI) [36] (if the clients worked) and the general self-efficacy scale (S-GSE) [37, 38]. The data collection process at baseline **(i)** started with that the assessment tools (Table 2) were sent to clients, who returned them by regular mail. Thereafter, a video meeting took place to complete the assessments and, through dialogues, support their self-reflections on their activities of everyday life and management strategies. The content of dialogues **(i)** was summarised in field notes. The assessment tools (Table 2) were again applied in a similar way combined with the dialogue at the four-month follow-up **(iv)** by the same researcher involved in the evaluation phase.

**Table 2. Overview and timeline of the data collection.**

| Data | (i) Baseline | (ii) Intervention process | (iii) Follow-up, one month | (iv) Follow-up, four months |
|---|---|---|---|---|
| *Background data of clients*: MFS[1], Sociodemographic data | X | | | |
| *Primary and secondary outcome assessments of clients*: POES[2], OBQ[3], Oval-pd[4], WAI[5], LiSat-11[6], GSE-10[7] | X | | | X |
| Field notes from researchers' meetings with clients: needs, management strategies expectations and accomplishment of SEE | X | X | X | X |
| Individual interviews client | | | X | X |
| Individual interviews OT | | | X | X |
| Registration forms of implementation of SEE | | X | | |

[1]Mental Fatigue Scale (MFS).

[2]Profile of occupational engagement (POES).

[3]Occupational Balance Questionnaire (OBQ).

[4]Concrete occupational value (Oval-pd).

[5]Work Ability (WAI).

[6]Satisfaction with life (Lisat-11).

[7]General self-efficacy (GSE-10).

Throughout the intervention process **(ii)**, the OTs completed registration forms of the implementation of SEE for each client after each module/session until the intervention ended. The forms covered how the technology used in the intervention functioned (health care platform, web program, video meetings) for both the client and the OT, whether the client adopted the intervention as intended, how the process evolved, and whether the education and intervention guide supported them in applying the various modules/sessions. If contacts were established with clients for any reason during the intervention process, filed notes were made to document the occasion. The researchers did not have access to the 1177 treatment platform or medical records.

At the one-month follow-up, which took place after the web program and establishment of the activity plan **(iii)**, qualitative semi-structured interviews [39] with clients (by IMB, MR) and their OTs (by IMB, MLL) were conducted by video meetings. The interview guides for the clients focused on their experiences with SEE´s content and delivery. The questions covered the clients' use of technology, barriers and facilitators related to technology use, as well as aspects of convenience, safety, and security. Questions were also asked about their experiences with each module/session and whether the SEE supported the development of self-management strategies. Finally, the clients were also asked how they established the activity plan with the OT and whether they would be able to implement it independently with the support of the OT. The interviews with the OTs focused on their experiences with providing SEE, including both technology use and the client´s intervention process. Furthermore, there was a need to further develop web-based education or the intervention guide.

At the follow-up four months after the intervention started **(iv)**, semi-structured interviews were again conducted with the clients (by IMB, MR) and their OTs (by IMB, MLL) through video meetings. This time, the clients' interview guides focused on the implementation of their activity plan, potential changes in everyday life and eventual needs for developing the intervention. The interview guide with the OTs covered how they had supported clients in implementing their activity plan/their change process, whether SEE changed their work and whether they had missed any content in SEE or support in the implementation. All the interviews **(iii, iv)** were audio recorded and transcribed verbatim.

One client did not complete the intervention. Therefore, after the fifth module in the web program, the one-month interview was conducted during a home visit. Additionally, observations of the use of the web program were documented in field notes. A complementary telephone interview with the client finalized the data collection.

## Data analysis

The data were analysed using pattern matching [25]. All data related to each participant (no. 1–4 clients, no. 5–6 OTs) constituted one unit of analysis. In the first step, all data (interviews, assessments, registration forms and field notes) were merged and summarised as a unit by the first author (IMB) to create a rich case description for each participant process (no. 1–6). In this, assessments were analysed with descriptive statistics. In the next step, the first and last authors (IMB, MLL) independently began to analyse each case description over time to identify patterns in the intervention process experiences, reflecting aspects of SEE feasibility and acceptability with a focus on content and delivery. The analysis continued with researchers identifying similarities and differences in patterns among the cases' descriptions. During this, each researcher identified that some of the participants' case descriptions reflected similar experiences of their process. These case descriptions were merged into a case, resulting in three cases, also constituting categories in the result. One client formed one case, three clients formed another case, and the OTs formed one case. In a final step, the evolving cases/categories and data were scrutinized by all authors and discussed repeatedly to ensure that they were grounded in data and to establish credibility [40]. During this process, the results were refined, and the final synthesis of the three cases/categories was completed.

## Results

The results consist of three categories: A) not being able to take on the internet-based intervention, B) being facilitated in the change process of daily life through the internet-based intervention, and C) providing a new internet-based intervention is a transition from ordinary practice. Each category includes two to four subcategories (Table 3), reflecting how aspects of SEE content and delivery were experienced differently in the three categories, as presented below. The presentation includes descriptions of whether changes were found in the outcome assessments for clients and detailed information of their descriptive values is given in Table 4.

**Table 3. Categories describing clients with stroke and their occupational therapists' experiences of the intervention process of SEE, reflecting feasibility and acceptability with a focus on content and delivery and experienced changes and outcomes of SEE.**

| Categories (Cases) | A) Not being able to take on the internet-based intervention | B) Being facilitated in change process of everyday life through the interne-based intervention | C) Providing a new internet-based intervention is a transition from ordinary practice |
|---|---|---|---|
| *Participants, no. 1–6* | *Client 1* | *Client 2, 3, 4* | *OT 5,6* |
| Subcategories: *(Aspects reflected in the subcategories)* | | | |
| *Acceptability and suitability of the content* | Wanting to get back to life as before | Experiencing a need for change in everyday life | |
| *Feasibility and acceptability of delivery* | Not being able to handle the intervention | Accessing support in a new way | Taking on a new way of delivering occupational therapy |
| *Feasibility and acceptability of content* | | Self-reflecting and adopting strategies for change | Supporting clients in a new way |
| *Experienced changes and outcomes of SEE* | | Having established a new approach to everyday life and experiencing changes | |

Table 4. The outcome of SEE in clients with stroke (n = 4).

| Participant Assessment, pre- and post intervention | Client 1 0/4 month | Client 2 0/4 month | Client 3 0/4 month | Client 4 0/4 month |
|---|---|---|---|---|
| Profile of occupational engagement[1] | 19/- | 32/32 | 25/35 | 31/35 |
| Occupational balance[2] | 13/- | 30/27 | 18/27 | 9/21 |
| Occupational values, total[3] | 50/- | 53/57 | 36/46 | 43/47 |
| Concrete occupational value[4] | 16/- | 18/17 | 12/18 | 15/15 |
| Symbolic occupational value[5] | 15/- | 17/19 | 12/13 | 14/16 |
| Self-reward occupational value[6] | 19/- | 18/21 | 12/15 | 14/16 |
| Self-efficacy[7] | 38/- | 33/33 | 25/35 | 33/32 |
| Satisfaction with life as whole[8] | 5/- | 5/5 | 4/6 | 3/4 |
| Satisfaction physical health[8] | 3/- | 4/4 | 4/6 | 3/4 |
| Satisfaction psychological health[8] | 4/ | 5/6 | 4/5 | 4/5 |
| Work ability[9] |  |  | 6/8 |  |

[1]POES, min-max score: 1–36 p.

[2]OBQ, min-max score: 0–33 p.

[3]Oval-pd, min-max score: 1–72 p.

[4]Oval-pd, min-max score: 1–36 p.

[5]Oval-pd, min-max score: 1–36 p.

[6]Oval-pd, min-max score: 1–36 p.

[7]GSE-10, min-max score: 10–40 p, a change of at least 2,97 has been suggested to identify a meaningful change.

[8]Lisat-11, Only three items from Lisat-11 were used, min-max score: 1–6 p.

[9]WAI, min-max score: 0–10 p.

## A) Not being able to take on the internet-based intervention

This category is based on one client (no. 1), who completed only five of the modules and terminated SEE after five months of attempts. Despite much support, the client recurrently struggled with managing technology as well as with grasping the content and purpose of SEE. The assessments (Table 4) revealed a need for extensive changes in daily activities, in accordance with the content of SEE, but according to the client's experiences, the expected change process never began because the internet format was not suitable.

**Wanting to get back to life as before.** Before entering SEE, the client expressed a need for support to resume lost activities and return to how life was before the stroke. The reason previous activities no longer were performed was unclear. The client reported being competent and able and needed support only with transportation, since driving a car was not possible. At the same time, the client´s description reflected that only a few activities were performed each day, although nothing really hindered engagement in more activities.

**Not being able to handle the intervention.** The interviews, as well as the registration form, showed that the client already had difficulties with managing the technology that was needed to obtain access to the intervention from the beginning. The difficulties were related to clicking on links to enter e-meetings and to understand the design of menus in the health care platform and the design of SEE. The difficulties with managing technology were not reported during inclusion, as the client described using the health care platform previously. The OT was described as helpful during the interviews, providing support in various ways to solve problems. However, the registration form showed that the client was not able to make use of the guidance given, and the problems persisted. This came to overshadow the overall experience of the intervention, and the client felt that it was not possible to participate. The client also postponed almost all modules and e-meetings with the OT due to personal matters. Interviews

and field notes from the observation in the client's home showed that the person had not been able to arrange a comfortable physical place or to plan and set aside enough time for the required sessions. The client's effort was put on issues other than focusing on the content of the intervention. Thus, the client's motivation decreased, as reflected in the last interview (five and a half months from the start): *"I have thought many times, I drop out! Because it has taken so much energy"* (client 1).

The interviews and registration forms showed that the client was not able to grasp or recount the content of the modules taken or that the overall goal of SEE was related to every-day life, despite having identified a need to change such aspects. As the client was not able to take on SEE, the decision to terminate the intervention and instead plan for other forms of rehabilitation were made after several dialogues among the OT, the client, and the researchers.

## B) Being facilitated in a change process of everyday life through the internet-based intervention

This category is based on three clients (no. 2–4) who all appreciated SEE and adopted a change process connected to their daily activities in various ways. The interviews revealed how the content and delivery of SEE supported them in adopting strategies that positively influenced their engagement in daily activities and life. Several assessments (Table 4) also showed client improvements at the four-month follow-up compared to preintervention.

**Experiencing a need for change in everyday life.** During the preintervention data collection, the clients expressed a need to change their daily activities to find a more satisfying every-day life in relation to their new condition. At the same time, they said they did not know how to make the changes and that they had difficulties identifying their problems. For example, one of the clients described a feeling of having a less meaningful everyday life and of doing nothing, despite each day being filled with activities. When asked about participating in SEE, one of the clients said, *"This is exactly what I'm thinking about [referring to the content of SEE], [my everyday life] is too boring, should I really live like this"* (client 4).

The assessments (Table 4) showed various limitations in the clients' patterns of daily activities and satisfaction with these. Furthermore, their satisfaction with life as a whole and their health varied. None of them used strategies systematically to overcome and prevent problems in daily activities.

**Accessing support in a new way.** The clients described that the internet-based format, the web program in the national health care platform and the e-meetings with the OT were easy to access. They had previous experiences of using the platform and of using e-meetings, but none of them had previously received an internet-based intervention through the platform. The internet-based format was seen as advantageous, as it saved both time and energy: *"This [the internet-based format] is much more comfortable, the hour that I would have spent on travelling, I can spend time on thinking and reflecting, so it feels like time is better spent"* (client 4).

It was also favourable to have the opportunity go back in the web program and repeat or check important content. For instance, see the video clips again to support the self-reflection in the change process. The clients described that it was as easy to meet in the e-meetings as in physical meetings, and the video enabled both visual and verbal confirmation during the dialogue. Thus, the e-meetings with the OT were experienced as supporting, safe and secure. One client also reflected that it was easier to share thoughts in an e-meeting from home than when meeting at the clinic: *"Well, the occupational therapist created a good atmosphere, it wasn't difficult to talk, maybe I even opened up more, because I feel safe here at home"* (client 4).

The clients said they planned so that they were able to sit on their own with a computer with a big screen, as they found that favourable. The sessions were occasionally interrupted,

e.g., an unexpected visit in the home. Both the interviews and the registrations form also revealed how technical issues disturbed the intervention on a few occasions. A lack of clinical routines regarding internet-based interventions resulted in inadequate reminders being sent, and a few unexplained errors occurred when the e-meetings were started. According to the clients, these were frustrating and took their focus away from the intervention to some extent until the problems were solved.

**Self-reflecting and adopting strategies for change.** The clients' experiences reflected how they, in various ways, took on a change process beginning from the initial modules. The register forms showed that all clients took the modules with the expected time interval of two to three weeks, even if one client expressed a desire to move forward more quickly and to get access to all the modules at once. Despite only reflecting to a limited extent on the content of the modules (interviews, register form), this client started to make changes and resume previous activities earlier than expected in the program. Furthermore, this client did not find any need to make an activity plan to continually establish and systematically work towards goals. Instead, the client had already established a change based on a new insight and acceptance of the situation, as reflected: *"It's rather important how you experience your life, if you realize that something has happened, that has changed [your everyday life] a lot, and you must let go of the comparisons with how it was before [the stroke]. Those thoughts have come from watching the films in the program"* (client 2).

The other two clients described how their own work with the modules, with support from the OT, helped them self-reflect on their patterns of daily activities, which gave them new insights important for their change processes. The modules' perspectives on daily activities became a tool that supported them in scrutinizing their everyday lives in a new manner, e.g., by scheduling daily activities over a week. Based on these insights, together with the OT, they designed an activity plan and identified strategies in daily activities that supported the planned changes. For example, their self-reflection led to insights that the perceived dissatisfaction in everyday life was related to low occupational balance, and by applying new strategies, they were able to change their engagement in new activities, as illustrated by the following citation: *"It [SEE] gives thoughts. . . I must say . . . it [SEE] guides you forward so to speak; you get an idea of what you have to change and what you can change"* (client 3).

The three clients' experiences reflected how they were supported to take an active role during the intervention and that the recurrent dialogue with the OT was crucial for their self-reflection and for moving the self-reflection process forward. As illustrated by one of the clients, *"It's motivating. . . . the OT will call me soon, so I must reflect [referring to content in the program]. . . if I do nothing, there can't be any change either. . . getting this push forward has been really good"* (client 4). The dialogues motivated them to continue with the change process, enabled them to see new possibilities, and encouraged them to try to do things differently. The recurrent dialogues were helpful, as they confirmed that their process progressed in the right direction but also confirmed each of them as a person with needs.

**Having established a new approach to everyday life and experiencing changes.** Four months from starting SEE, the interviews and the assessment together reflect how the three participants experienced positive changes in everyday life by having established a new way to manage activities. The assessments (Table 4) revealed different improvements in the three clients. For example, the self-reported occupational values (Oval-pd) improved for all, especially the symbolic and self-rewarding values. The Occupational engagement (POES) and satisfaction with occupational balance (OBQ-11) increased for two of the clients (no. 3, 4). An increase in satisfaction with psychological health was reported by all three clients and two (No., 3,4) of them also reported an increase in satisfaction with physical health and life as whole. The self-efficacy (GSE-10) remained stable except for one client (No.,3) who perceived

an increase and, also, perceived an improved work ability (WAI). Furthermore, the interviews reflected the significance of SEE: *"I'm glad I participated. . . it's been good to take one step at a time. . . I like this format. It has been useful and has helped me to feel better psychologically"* (client 4).

The clients' experiences from the interviews reflected how they had developed an understanding of the effects their overall pattern of activities have on their well-being. With the support of SEE, they created strategies in the form of routines or overall approaches on how to manage everyday life, as well as a readiness to manage the future. It was clear that not all clients thought of their new established routines and approaches in terms of strategies. Instead, they described how they thought and acted differently in everyday life. One of the clients said, *"Yes, the big change is to do [tasks] as long as it's fun instead of doing until it's done so to speak. There have been thoughts about it [before], to take breaks and rest, so that's an overall strategy so to speak"* (client 2).

All client descriptions reflected changed patterns of activities over the day/week for a variety of activities, and a feeling of control and occupational balance was described. Those (no. 3, 4) who had established the activity plan said the plan had guided them forward and continued to serve as a support and a reminder of what is important for them to feel well: *"I think the plan has worked well, I don't put off what needs to be done anymore, I am better on getting things done. I have done a lot of things that have hanged over me"* (client 3).

## C) Providing a new internet-based intervention is a transition from ordinary practice

This category is based on two OTs (no. 5,6) who described how providing SEE was experienced as a transition from ordinary practice and that the content and delivery was feasible and acceptable in their clinical practice.

**Taking on a new way of delivering occupational therapy.** In the interviews, the OTs described that they had no previous experiences with internet-based interventions. However, they said that the educational program (both the SEE content and how to manage SEE in the health care platform) as well as the SEE intervention guide were sufficiently supportive and prepared them for implementation with clients. The intervention guide was experienced as thorough and easy to adhere to for each module and, in combination with access to supervision when needed, made them feel confident when they were new users. The interviews and the registration forms confirmed that SEE was implemented as intended, even if everything was experienced as new and energy consuming.

The OTs' experiences delivering SEE showed that they needed to change their work processes and deal with several new challenges while being faced with the benefits of working with an internet-based intervention. One challenge was the work environment not being adapted for client interventions through the internet, which demanded planning to obtain access to individual rooms and computers that could ensure privacy. Another challenge was the increased demand for digital competence that had not been requested previously in their daily work, e.g., sharing screens with clients when establishing an activity plan. The challenges they faced and how they managed these depended to a large extent on their ability to use technology outside work. Another challenge was the lack of routines in the organization and limited access to support when technical issues arose. Taken together, these challenges caused stress before and during e-meetings with clients, which negatively influenced the focus on the client's process: *"The first meeting was a bit uncomfortable, because I was stressed by the technology and didn't get into the meeting. I felt that it made me unfocused, I wasn't pleased at all after that session. Other times it has worked well when we have met. . . In addition, the program itself is new,*

*no matter how much you prepare, it is a new environment or whatever I should call it, to work with for me*" (OT 6).

The OTs experienced remote difficulties with supporting clients when technical issues arose. For instance, instable internet access or entering e-meetings, especially if the clients were not used to manage such issues. Unexpected visits in the client's home during e-meetings were difficult to handle, and it interrupted the ongoing dialogues. Despite these challenges, the OTs described the benefits of e-meetings as opening up a new type of dialogue that became richer: *"I think that. . . . Well sure you might get a different connection when you meet face to face and sit across each other at a table, but I don't know if I could have. . . asked exactly as many questions about things or absorbed all the information as I did when we met on Skype. I think it worked out great"* (OT 5).

The OTs described how the design of the modules in SEE helped the clients achieve new knowledge, which also prepared clients for the e-meetings. This was favourable for the dialogues and the clients' change process. Furthermore, time could be used more efficiently: *"It's also timesaving,. . . The program is spread over quite a few hours and the patients get a lot of knowledge and information, but a lot of the [patient's] work is done independently. I [as an OT] are not there all the time"* (OT 5).

**Supporting clients in a new way.**   The OTs' experiences reflected that they did not have the same role in the intervention or as much control as they were used to in their ordinary practice, as the clients immediately started to work on their own with self-reflection based on the different modules: *"You are used to guide them hands-on or giving tips and advice in the situation. It [SEE] is a slightly different way of working from the way I am used to working, . . . that the patients drive their own process, and I am there to support and guide them, to make them reflect"* (OT 6). Between the modules with e-meetings, OTs were unsure how far the clients had come in their self-reflection. They also discovered how the clients' levels of self-reflection varied, and in combination with their various needs in daily activities, the OTs found it both challenging and stimulating to work with SEE. Another aspect discussed was the importance of being well prepared for various scenarios to guide the client forward. The OTs' narrations about some clients showed how therapists during the dialogues with the client could become unsure of how much he/she had reflected over daily activities, and how this could be managed during the dialogues. To facilitate the clients' self-reflections, therapists adhered to the intervention guide and replayed the clients' performances on tasks in the web program.

If a client did not follow the expected process, e.g., chose not to take the modules as intended, the OT became unsure of how to handle the situation, as this was not described in the intervention guide. Uncertainty when the activity plan was established was also expressed, as they were not used to formulating goal plans, and such information was not included in the SEE intervention guide. Overall, their experiences of providing the SEE intervention process in its current design were positive and viewed as a new sustainable way of working as an OT: *"That the patient takes quite a lot of self-responsibility right away when the intervention starts and that you [as an OT] are there more as a support. . . I think it's a good approach if you want to reach a sustainable change"* (OT 5).

## Discussion

The aim of this study was to explore and describe how clients with stroke and their OTs experienced the intervention process of SEE and whether SEE has the potential to promote an active everyday life. The first test of the intervention process indicates that the content and delivery of SEE can be feasible and acceptable both for clients and OTs. The findings suggest that SEE has the potential to support clients' self-reflections and their adoption of strategies that

influence engagement in daily activities and satisfaction with life in various ways. Additionally, the educational program and the intervention guide of SEE together with supervision have the potential to support OTs in guiding clients forward in their change process. However, the results need to be considered in the light of the case study design and the fact that this initial clinical testing of SEE was limited to one rehabilitation setting. The result indicates, in line with recent research [21, 41, 42], the potential to develop stroke rehabilitation by applying various internet-based solutions. In contrast to other new (occupational therapy based) interventions [42–44] that foremost incorporate internet-based solutions as a component of the program, SEE is completely internet-based throughout the interventions process. The results can therefore provide important knowledge when other internet-based interventions are developed. As other stroke interventions foremost focus on needs in the earlier phases after stroke and on restoring functioning [45]. The results add insights in how the change process people need to go through to adapt to an active everyday life on new conditions can be targeted in programs in the later phases of rehabilitation. Moreover, our results indicate that after some minor adjustments, the evaluation of SEE is ready to move to the next phase, i.e., the feasibility phase including research with larger samples and a quantitative approach, in accordance with the guidelines for the development of complex interventions [24].

All four clients perceived that the content of SEE was relevant for their needs (Table 3). This indicates that the SEEs' focus, and content match clients' rehabilitation needs that were previously identified through reviews of research during the development of the intervention [14, 46–48]. However, for one of the clients (category A), it was found that the internet-based format of delivery was not suitable. The results indicate that this can be difficult to detect through dialogues, especially if a client have not tested the format before and if they are not fully aware of their abilities to use the health care platform. It can also be difficult to detect whether a client can take on an active role in their change processes prior to starting the intervention. Playing an active role in their change processes is also associated with the clients' motivations being changeable [27]. If something in the intervention becomes unmanageable, such as technology, motivation be lost. Consequently, it is as important to want to change as it is to be ready and able to change (e.g., use needed technology) [49]. When difficulties occur, as for the client in category A, it is important to provide support sensitive to why SEE is not working or to offer another form of rehabilitation. In the future, assessments of the ability to use technology [50–52] might be a way to identify whether the internet-based format is suitable for a client.

As it was the first time that the included clients and OTs participated in an internet-based intervention, the feasibility and acceptability of SEE delivery were influenced by the participants' ability to manage the technology involved as well as environmental aspects. The results revealed the importance of reviewing whether the physical, social, and organizational work environment [53] supports the transition to an internet-based service by providing, e.g., individual rooms, technological equipment, education, support, and digital solutions for time management. This is in line with recent discussions [54, 55] of urgent issues to maximize the potential of the digital transformation of health care. For both parties, it was important to prepare for the modules and e-meetings to optimise their outcomes, e.g., by setting aside enough time and choosing a place to avoid interruptions. Based on these experiences, some small amendments related to SEE delivery have been made. The intervention guide as well as the client information now includes recommendations on how to prepare for the intervention sessions. Routine documents have been established to support OT, e.g., in how to contact clients digitally and how to share screens with clients when the activity plan is established.

The results from the three clients in category B and their OTs show how technology, when working as intended, can be perceived as advantageous over physical face-to-face meetings. The three clients described, in line with research [56], how they saved energy and time by not having to transport themselves to the clinic. Instead, valuable time could be spent on processing and repeating the content of the intervention. From the two OTs' perspective, this improved their ability to support clients' change process. Our results indicate that the therapeutic relationship and dialogue can be enhanced by SEE, where OTs can spend more time supporting clients in processing content rather than providing information. This confirms the value of applying the flipped-classroom methodology [57] as a means for supporting distance-based learning and processes in health care services.

The experiences of the three clients in category B reflected how SEEs' content and delivery fostered them to take responsibility and lead their change processes. Based on the knowledge gained from the modules, their self-reflections of engagement in daily activities were developed, and new strategies were adopted. This indicates, similar to occupational therapy models [58–60] and interventions with a similar focus [61–63], the advantages of supporting peoples' self-reflections on patterns, balance and values in activities in everyday life. Additionally, the results indicate in accordance with occupational therapy models [58–60] that focusing on engagement in daily activities has advantages both as a therapeutic means and as a goal in the change process. Facilitating people's management resources and self- initiated strategies in activities of everyday [64] can be seen as important due to the changes taking place in the health care system [65, 66], here people is expected to take responsibility and act proactively to promote their own health. As existing occupational therapy models [60, 67] focuses on professional strategies rather than on clients own self- initiated strategies, future intervention research as of SEE can contribute to model development.

The fact that the clients were responsible for their own change processes and self-reflections allowed their change processes to develop in different ways. This was described by the two OTs as new and challenging for supporting clients. On the other hand, in line with the literature on motivational interviewing [49], this was perceived by the OTs to strengthen the sustainability of the intervention in the long term. As SEE involves a transition from the OT´s ordinary way of working, both in terms of content and delivery, education and supervision are needed. Based on the participants' experiences, the educational program, the intervention guide, and the structure for supervision were updated. A completed activity plan was included as an example in the intervention guide. Moreover, to support the clients' preparation prior to establishing the activity plan, information sheets supporting self-reflection have been developed.

The results indicate that SEE, when implemented as intended, has the potential to change everyday life (engagement in daily activities, occupational balance, occupational value, and aspects of life satisfaction). Notably, client 2 in category B, who chose not to establish an activity plan and continue the process without guidance from the OT, did not achieve positive changes that were reflected in the assessments to the same extent as clients 3–4, who completed the whole SEE process. However, the qualitative descriptions (from both client 2 and the OT) reflected that those changes in everyday life had been implemented successfully even if the changes were reflected to a limited extent in the assessments. After improvements in SEE, feasibility studies are needed to further explore potential effects as well as to deepen the understanding of components influencing the intervention processes and their outcomes from both a client and an OT perspective. Future research also needs to investigate resource utilization as time consumption in relation to outcomes of SEE and how the change of the OTs practice (using SEE) can be integrated with the practice of other professionals in interdisciplinary teams.

## Methodological considerations

The qualitative descriptive case study design, including both clients and their OT, combining multiple data collection methods resulted in rich descriptions from various aspects of each case that enhance the understanding of credibility and transferability [40] of the results. Even if this first clinical test is based on a small sample, i.e., four clients and two OTs, the results contributed with information that led to several improvements in SEE content and delivery. From an ethical and resources utilization perspectives, it can be favorable to limit the initial testing to a few cases prior to implementing feasibility studies with larger samples. It is possible that the credibility [40] of the data might have been limited since the participants knew the researchers who collected the data and were a part of the team who developed and evaluated SEE. This could potentially have hampered them from talking freely about their experiences, even if they were repeatedly told that positive and negative experiences were equally important to developing SEE. The client participants were consecutively invited from a list with a potential to ensure we reach a variation of participants. When four had agreed to participate the inclusion needed to stop due the COVID-19 outbreak. This resulted in a rather homogeneous sample even if the number of women and men included in both participant groups were equal. If more participants had been included, they would probably have contributed with additional perspectives. However, the depth of data of the cases comprised were judged to be sufficiently rich to ensure the quality of the first clinical test. Further, it was beyond the scope of this case study to focus on differences related to the participant's background data. In forthcoming feasibility studies, it will be important to ensure variation of participants related to background data as age and, also, to include several rehabilitation settings. The authors, all of whom were OTs, had various experiences of clinical rehabilitation, distance-based learning, and research. To ensure trustworthiness [40], the authors served different roles with their various expertise to support each other to remain open during the analysis process when the results evolved.

## Conclusion

In conclusion, the first test of the intervention process indicates that the content and delivery of SEE can be feasible and acceptable to both clients and OTs. The findings suggest that SEE has the potential to support clients' self-reflections and their adoption of strategies that influence engagement in daily activities and satisfaction with life in various ways. This study identified several minor improvements in content and delivery needed to strengthen the implementation of SEE intervention processes in the forthcoming feasibility phase.

## Acknowledgments

The authors are grateful to the participants who took on the new intern-based intervention and shared their experiences. We are also grateful to the management team for Paramedicine, Sunderby Hospital, Region Norrbotten, for enabling the implementation of SEE.

## Author Contributions

**Conceptualization:** Anneli Nyman, Eva Månsson Lexell, Maria Larsson-Lund.

**Data curation:** Ida-Maria Barcheus, Maria Ranner, Maria Larsson-Lund.

**Formal analysis:** Ida-Maria Barcheus, Maria Ranner, Anneli Nyman, Eva Månsson Lexell, Maria Larsson-Lund.

**Funding acquisition:** Anneli Nyman, Eva Månsson Lexell, Maria Larsson-Lund.

**Investigation:** Ida-Maria Barcheus, Maria Ranner, Maria Larsson-Lund.

**Methodology:** Ida-Maria Barcheus, Maria Ranner, Anneli Nyman, Eva Månsson Lexell, Maria Larsson-Lund.

**Project administration:** Maria Larsson-Lund.

**Resources:** Ida-Maria Barcheus, Maria Larsson-Lund.

**Supervision:** Maria Ranner, Maria Larsson-Lund.

**Validation:** Ida-Maria Barcheus, Maria Ranner, Anneli Nyman, Eva Månsson Lexell, Maria Larsson-Lund.

**Visualization:** Ida-Maria Barcheus, Maria Larsson-Lund.

**Writing – original draft:** Ida-Maria Barcheus, Maria Larsson-Lund.

**Writing – review & editing:** Ida-Maria Barcheus, Maria Ranner, Anneli Nyman, Eva Månsson Lexell, Maria Larsson-Lund.

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
