## [Decision Letter · Decision Letter 0]

20 Jul 2022

PONE-D-22-04745Developing and testing the feasibility of a new internet-based intervention– a case study of people with stroke and occupational therapistsPLOS ONE

Dear Dr. Barcheus,

Thank you for submitting your manuscript to PLOS ONE. After careful consideration, we feel that it has merit but does not fully meet PLOS ONE’s publication criteria as it currently stands. Therefore, we invite you to submit a revised version of the manuscript that addresses the points raised during the review process.

Please note that we have only been able to secure a single reviewer to assess your manuscript. We are issuing a decision on your manuscript at this point to prevent further delays in the evaluation of your manuscript. Please be aware that the editor who handles your revised manuscript might find it necessary to invite additional reviewers to assess this work once the revised manuscript is submitted. However, we will aim to proceed on the basis of this single review if possible. Their comments are attached below. The reviewer suggests some clarifications on phrases and definitions used and has questions regarding the data analysis. Could you please revise the manuscript to address all their queries?

We look forward to receiving your revised manuscript.

Kind regards,

Thomas Tischer

Staff Editor

PLOS ONE

Journal Requirements:

This study was supported by grants from the Kamprad Family Foundation for Entrepreneurship, Research & Charity. The funder had no role in the study.

6. Please include a copy of Table 4 which you refer to in your text on page 12.

Additional Editor Comments:

It is appreciated that some limitations of the study are discussed, if possible please include a comment about the small sample size and the limited age range of the participants.

Reviewers' comments:

Reviewer's Responses to Questions

**Comments to the Author**

1. Is the manuscript technically sound, and do the data support the conclusions?

Reviewer #1: Partly

2. Has the statistical analysis been performed appropriately and rigorously? 

Reviewer #1: N/A

3. Have the authors made all data underlying the findings in their manuscript fully available?

Reviewer #1: Yes

4. Is the manuscript presented in an intelligible fashion and written in standard English?

Reviewer #1: Yes

5. Review Comments to the Author

Reviewer #1: Thank you for having the opportunity to review the manuscript Developing and testing the feasibility of a new internet-based intervention– a case.

Overall

The paper presents an interesting research study aiming to enhance knowledge on developing stroke rehabilitation within the research field of occupational therapy. However, there are some concerns regarding the presentation of the study.

The title embraces the content of the study. The abstract includes relevant information, and the key words are not already used in the title.

The introduction explains the background of the study grounded in updated literature and the rationale of the study states the significance for occupational therapy.

Page 3

- lines 69 -70 – please insert a well-known definition on occupational balance and one or more references.

- Lines 75- 76 – please rephrase the sentence and include a well-known definition of self-management.

- Line 77 – please clarify the term “facility” in this context.

- Line 78 – please give examples of common unfulfilled needs

- Line 79 – please develop the argument for a new mode of service. Why will a more stable delivery of existing rehabilitation services not be sufficient?

Page 4

- Line 89 – please define the concept of empowerment and provide relevant references

The methods and materials used are overall well presented and argued. Considerations are taken to provide appropriate and trustworthy procedures and ethical considerations are accounted for.

I recommend further clarity of the inclusion criteria:

- moderate disability or good recovery

- motivated to change their activities in everyday life

And an elaboration on how the screening for inclusion was undertaken as the inclusion criteria seem quite unclear.

Moreover, please develop the argumentation for data collection tools related to the data collection periods – and associate the data collection to the concept of empowerment as well as activities of daily life and management strategies.

In this section the authors may also want to state and reflect on their preunderstanding.

The result section present findings relevant to the aim of the study. The table adequately enhance the presentation of the material.

In table 3 delivery is mentioned twice. How come?

Page13

- line 271 – it is unclear who provide the narrations

Page 15

- lines 330 - erase small line

The discussion section considers the findings in view of previously reported research and avenues for future research is suggested.

Page 21

- please elaborate on the feasibility of SEE in the clinical setting of the OT – how time consuming is the introduction to and the use of the SEE compared to usual practice? How did the change in the OT practice affect the practice of other professionals in the interdisciplinary teams?

The tentative reflections on the findings may be in want of more nuances and critical reflections. How does these findings differ from or develop already well-known research? Please develop and elaborate on the statements concerning the relevance for occupational therapy practice as the findings are based on very few persons and apparently one single rehabilitation setting – e.g., please expand the discussion on, why variation did not enter into the inclusion as this is an issue in relation to the quality of the research, and how these findings relate to the concept of empowerment and occupational therapy theories in the context of stroke rehabilitation and occupational therapy.

Reflection on need for future research and the methodological considerations present very important reflections that may well have been elaborated, included in, and qualified the discussion.

6. PLOS authors have the option to publish the peer review history of their article (what does this mean?). If published, this will include your full peer review and any attached files.

Reviewer #1: No

---

## [Author Response · Author response to Decision Letter 0]

23 Sep 2022

Response to Editor and Reviewer have been provided in an attached document labelled "Response to Reviewers" where point by point responses are provided for clear reading. We hope it will work well for you.

---

## [Decision Letter · Decision Letter 1]

11 May 2023

PONE-D-22-04745R1Developing and testing the feasibility of a new internet-based intervention– a case study of people with stroke and occupational therapistsPLOS ONE

Dear Dr. Barcheus,

Thank you for submitting your manuscript to PLOS ONE. After careful consideration, we feel that it has merit but does not fully meet PLOS ONE’s publication criteria as it currently stands. Therefore, we invite you to submit a revised version of the manuscript that addresses the points raised during the review process.

Please note that your revision is evaluated by a new reviewer in addition to the previous review. We usually do not invite different sets of reviewers for revised manuscripts. In this case, however, the editor who handles your revision find it is necessary to do so. Please see the new comments below.

We look forward to receiving your revised manuscript.

Kind regards,

Jianhong Zhou

Staff Editor

PLOS ONE

Reviewers' comments:

Reviewer's Responses to Questions

**Comments to the Author**

1. If the authors have adequately addressed your comments raised in a previous round of review and you feel that this manuscript is now acceptable for publication, you may indicate that here to bypass the “Comments to the Author” section, enter your conflict of interest statement in the “Confidential to Editor” section, and submit your "Accept" recommendation.

Reviewer #1: All comments have been addressed

Reviewer #2: (No Response)

2. Is the manuscript technically sound, and do the data support the conclusions?

Reviewer #1: Yes

Reviewer #2: Partly

3. Has the statistical analysis been performed appropriately and rigorously? 

Reviewer #1: N/A

Reviewer #2: No

4. Have the authors made all data underlying the findings in their manuscript fully available?

Reviewer #1: Yes

Reviewer #2: No

5. Is the manuscript presented in an intelligible fashion and written in standard English?

Reviewer #1: Yes

Reviewer #2: Yes

6. Review Comments to the Author

Reviewer #1: The paper presents an interesting research study aiming to enhance knowledge on developing stroke rehabilitation within the research field of occupational therapy. The manuscript is a technically sound piece of scientific research. Methods have been conducted and reported rigorously. Discussions and conclusion are appropriately based on the results presented. All previous comments has been addressed. Hence, this manuscript is now acceptable for publication.

Reviewer #2: Abstract

Introduction need to be concrete showing the needs for this study

Method lack of information how the data is collected, who are the participants and number of participants, demo profile of the participants, how the data were analyzed

Findings - poorly written

Results

-should include demo profile of the participants, level of education, severity of stroke

The authors should consider quantitative survey instead of using qualitative case study methodology for this study.

The authors should focus perhaps user / patients responds and OT feedback/respond in different study.

7. PLOS authors have the option to publish the peer review history of their article (what does this mean?). If published, this will include your full peer review and any attached files.

Reviewer #1: No

Reviewer #2: No

---

## [Author Response · Author response to Decision Letter 1]

17 May 2023

Response to Editor and Reviewer have been provided in an attached document labelled "Response to Editor and Reviewers" where point by point responses are provided for clear reading. To editor we have attached comments in cover letter. We hope it will work well for you.

---

## [Decision Letter · Decision Letter 2]

15 Nov 2023

PONE-D-22-04745R2Developing and testing the feasibility of a new internet-based intervention– a case study of people with stroke and occupational therapistsPLOS ONE

Dear Dr. Barcheus,

Thank you for submitting your manuscript to PLOS ONE. After careful consideration, we feel that it has merit but does not fully meet PLOS ONE’s publication criteria as it currently stands. Therefore, we invite you to submit a revised version of the manuscript that addresses the points raised during the review process.

The manuscript has been evaluated by one reviewer, and their comments are available below.

The reviewers has raised a number of concerns that need attention, could you please revise the manuscript to carefully address the concerns raised?

We look forward to receiving your revised manuscript.

Kind regards,

Vanessa Carels

Staff Editor

PLOS ONE

Journal Requirements:

Reviewers' comments:

Reviewer's Responses to Questions

**Comments to the Author**

1. If the authors have adequately addressed your comments raised in a previous round of review and you feel that this manuscript is now acceptable for publication, you may indicate that here to bypass the “Comments to the Author” section, enter your conflict of interest statement in the “Confidential to Editor” section, and submit your "Accept" recommendation.

Reviewer #3: All comments have been addressed

2. Is the manuscript technically sound, and do the data support the conclusions?

Reviewer #3: Partly

3. Has the statistical analysis been performed appropriately and rigorously? 

Reviewer #3: N/A

4. Have the authors made all data underlying the findings in their manuscript fully available?

Reviewer #3: Yes

5. Is the manuscript presented in an intelligible fashion and written in standard English?

Reviewer #3: Yes

6. Review Comments to the Author

Reviewer #3: This paper investigated the PLOS ONE manuscript "Developing and testing the feasibility of a new internet-based intervention– a case study of people with stroke and occupational therapists." I think this paper needs minor revision to improve the rationale and gap in knowledge of the study. Specific comments for each section are below:

Introduction

Page 3, Line79: Before going to “A new mode of service”, please give the rationale or gap in knowledge why internet-based intervention could be fulfilled in the future stroke service, it would be helpful to insert addition in the one or two sentences.

Page 4, Line 91: the MRC guideline needs to give a full name.

Methods

Page 7, Line 153: What is the Swedish 1177? Is it referring to a health care advice phone or hotline? Please clarify this point.

Page 7, Line 163: It would be helpful to provide a brief of web program in which it consists of 8 modules. Please clarify this point that help readers to understand the mechanism of internet-based intervention.

Result

Page 12, Line 264: authors mentioned Table 4, where is table 4?

Page 14, Line 303: authors mentioned Table 4, where is table 4?

Page 15, Line 313: authors mentioned Table 4, where is table 4?

Page 17, Line 378: authors mentioned Table 4, where is table 4?

Page 17, Line 379-381: It was the only Oval-pd and OBQ to support the statement "Having established a new approach to everyday life and experiencing changes". Is there a section for results from POES, WAI, Lisat-11, and GSE-10 to support this? You mentioned four months of follow-up in Table 2 (iv). There is a point in this paper that would strengthen it and make it more understandable to the reader.

Discussion

Page 24, Line 545-549: “Client 2 in category B, who chose not to establish an activity plan and continue the process without guidance from the OT, did not achieve positive changes that were reflected in the assessments. However, the qualitative descriptions (from both client 2 and the OT) reflected that those changes in everyday life had been implemented successfully.” It is likely that these two messages contradict each other. Please clarify this point.

The revisions you suggest will lead to a significant improvement in the quality and impact of your paper. It would be great to see your revised manuscript, and I thank you for your contribution.

Best wishes,

Reviewer

7. PLOS authors have the option to publish the peer review history of their article (what does this mean?). If published, this will include your full peer review and any attached files.

Reviewer #3: No

---

## [Author Response · Author response to Decision Letter 2]

27 Nov 2023

Response to the Peer Review Operations Manager, Staff Editor, and the reviewer for the revised manuscript submission (PONE-D-22-04745R2) November 2023 

Response to Editor and Reviewers 

Journal Requirements:

Please review your reference list to ensure that it is complete and correct. If you have cited papers that have been retracted, please include the rationale for doing so in the manuscript text or remove these references and replace them with relevant current references. Any changes to the reference list should be mentioned in the rebuttal letter that accompanies your revised manuscript. If you need to cite a retracted article, indicate the article’s retracted status in the References list and also include a citation and full reference for the retraction notice.

Reply: The reference list has been updated with full details. Similarly, three new references (no.15-18) have been added following clarifications in the introduction. 

Review Comments to the Author:

Reviewer #3: This paper investigated the PLOS ONE manuscript "Developing and testing the feasibility of a new internet-based intervention– a case study of people with stroke and occupational therapists." I think this paper needs minor revision to improve the rationale and gap in knowledge of the study. Specific comments for each section are below:

Reply: Thank you for the overall positive response and that only minor revisions remain.

Introduction

Page 3, Line79: Before going to “A new mode of service”, please give the rationale or gap in knowledge why internet-based intervention could be fulfilled in the future stroke service, it would be helpful to insert addition in the one or two sentences.

Reply: Thank you for bringing this to our attention, we have now developed the rationale for the study. 

Page 4, Line 91: the MRC guideline needs to give a full name.

Methods

Page 7, Line 153: What is the Swedish 1177? Is it referring to a health care advice phone or hotline? Please clarify this point.

Page 7, Line 163: It would be helpful to provide a brief of web program in which it consists of 8 modules. Please clarify this point that help readers to understand the mechanism of internet-based intervention. 

Reply: Thank you for bringing this to our attention, clarifications have been made as requested.

Result

Page 12, Line 264: authors mentioned Table 4, where is table 4?

Page 14, Line 303: authors mentioned Table 4, where is table 4?

Page 15, Line 313: authors mentioned Table 4, where is table 4?

Page 17, Line 378: authors mentioned Table 4, where is table 4?

Reply: Table 4 is already in the manuscript. The Table can be found directly when it is mentioned the first time in the results. It is placed under Table 3.

Page 17, Line 379-381: It was the only Oval-pd and OBQ to support the statement "Having established a new approach to everyday life and experiencing changes". Is there a section for results from POES, WAI, Lisat-11, and GSE-10 to support this? You mentioned four months of follow-up in Table 2 (iv). There is a point in this paper that would strengthen it and make it more understandable to the reader. 

Reply: We have clarified that the (results in the) category are based on data both from interviews and assessments. We have also made adjustments in the text and added a sentence about the other assessment not previously mentioned. 

Discussion

Page 24, Line 545-549: “Client 2 in category B, who chose not to establish an activity plan and continue the process without guidance from the OT, did not achieve positive changes that were reflected in the assessments. However, the qualitative descriptions (from both client 2 and the OT) reflected that those changes in everyday life had been implemented successfully.” It is likely that these two messages contradict each other. Please clarify this point.

Reply: The requested clarification has been made.

---

## [Decision Letter · Decision Letter 3]

12 Dec 2023

Developing and testing the feasibility of a new internet-based intervention– a case study of people with stroke and occupational therapists

PONE-D-22-04745R3

Dear Dr. Barcheus,

We’re pleased to inform you that your manuscript has been judged scientifically suitable for publication and will be formally accepted for publication once it meets all outstanding technical requirements.

Kind regards,

Fatma Refaat Ahmed, Ph.D.

Academic Editor

PLOS ONE

Additional Editor Comments (optional):

Reviewers' comments:

Reviewer's Responses to Questions

**Comments to the Author**

1. If the authors have adequately addressed your comments raised in a previous round of review and you feel that this manuscript is now acceptable for publication, you may indicate that here to bypass the “Comments to the Author” section, enter your conflict of interest statement in the “Confidential to Editor” section, and submit your "Accept" recommendation.

Reviewer #1: (No Response)

2. Is the manuscript technically sound, and do the data support the conclusions?

Reviewer #1: (No Response)

3. Has the statistical analysis been performed appropriately and rigorously? 

Reviewer #1: (No Response)

4. Have the authors made all data underlying the findings in their manuscript fully available?

Reviewer #1: (No Response)

5. Is the manuscript presented in an intelligible fashion and written in standard English?

Reviewer #1: (No Response)

6. Review Comments to the Author

Reviewer #1: (No Response)

7. PLOS authors have the option to publish the peer review history of their article (what does this mean?). If published, this will include your full peer review and any attached files.

Reviewer #1: No

---

## [Editor Report · Acceptance letter]

18 Dec 2023

PONE-D-22-04745R3 

PLOS ONE

Dear Dr. Barcheus, 

I'm pleased to inform you that your manuscript has been deemed suitable for publication in PLOS ONE. Congratulations! Your manuscript is now being handed over to our production team.

Kind regards, 

on behalf of

Dr. Fatma Refaat Ahmed 

Academic Editor

PLOS ONE